# Time Series Anomaly Detection using Reconstruction and RBF Similarity Scores

## Abstract

Anomaly detection in time series data is pivotal across various domains. The inherent challenge of scarce labeled data for anomaly detection has increased the attention toward unsupervised learning methods, in particular autoencoders and variations thereof. While these unsupervised approaches have shown promise, those that solely rely on reconstruction error often miss subtle anomalies, especially in high-dimensional or multivariate datasets. Motivated by this challenge, we introduce a novel approach that utilizes a layer of Radial Basis Function (RBF) neurons within the deep learning architectures. This RBF layer fits a nonparametric density in the hidden representation. When the neural network is trained on (predominantly) normal data, then a high RBF output indicates a high density, which in turn implies a high similarity with the normal data. Combining the RBF similarity score with the reconstruction error results in a unique anomaly score that we named the *SimRec score*. While our method can be adapted to a wide range of architectures, we focus on LSTM and Transformer models. We evaluate our approach on three real-world benchmark datasets, with results indicating significant improvements over the baselines. Our findings underscore the potential of the SimRec score in capturing subtle anomalies that might be overlooked by scores based on reconstruction error alone, offering a more robust and comprehensive solution for anomaly detection in time series data.

## 1 Introduction

Anomalies in time series data, such as unexpected deviations or patterns, can signify critical issues in various application domains, ranging from fraudulent financial transactions to life-threatening health conditions. Hence, accurate anomaly detection is important. Given the rarity of anomalies and, thus, the lack of sufficient labeled data, fully supervised prediction methods are less suited. Therefore, unsupervised learning methods have gained increasing attention. These methods offer advantages over supervised approaches as they do not rely on explicitly labeled examples of the anomalies. This makes them more adaptable to the complex and rare nature of anomalies, as well as better suited for detecting unknown or unexpected anomalies Chandola et al. (2009); Ghorbani et al. (2023).

Various classic unsupervised anomaly detection techniques, such as clustering-based methods like One-Class Support Vector Machine (OC-SVM) Schölkopf et al. (1999) or Support Vector Data Description (SVDD) Tax & Duin (2004), as well as density-estimation approaches, like Local Outlier Factor (LOF) Breunig et al. (2000), have been widely used in different domains. While these methods can be adapted to handle time series data, they face inherent challenges due to the temporal dependencies, high dimensionality, and the complex generalization requirements of such data Mejri et al. (2022). Recent advances in deep learning have shown promising results for anomaly detection in time series data Choi et al. (2021). Architectures such as Transformers, Long Short Term Memory (LSTM), and LSTM-autoencoders are capable of capturing temporal patterns Tuli et al. (2022); Hundman et al. (2018); Audibert et al. (2020a). These models can automatically learn and extract hierarchical and non-linear features, which enables them to effectively handle the challenges of temporal dependencies and high dimensionality in time series data. These models are originally constructed for prediction and reconstruction tasks, but can be extended to perform anomaly detection.

Building on these advancements, a variety of effective anomaly detection methods have been developed Zhou et al. (2019); Li et al. (2021); Park et al. (2018); Ding et al. (2023). When categorizing these studies based on the anomaly criteria score, most of them are centered around Reconstruction Error (RE). For instance, OmnyAnomaly from Su et al. (2019) proposes a stochastic recurrent neural network model to capture uncertainties in time series data using reconstruction probability. The USAD method by Audibert et al. (2020b) utilizes a unique architecture that combines the strengths of autoencoders to identify subtle and complex anomalies using the RE metric. Furthermore, STAD-GAN from Zhang et al. (2023) introduces an innovative self-training Generative Adversarial Network approach, enhancing robustness against anomalies using the RE metric. However, it is important to note that not all these studies use pure RE as the sole anomaly score.

Relying on the reconstruction error during the unsupervised learning step poses some challenges for anomaly detection, especially in high-dimensional or multivariate time series data. One primary concern is the smoothing effect, where models, particularly those based on deep learning architectures, tend to average out anomalies during the reconstruction process. This effect can lead to a reduced sensitivity, causing subtle anomalies to be overlooked. This challenge is exacerbated in multivariate time series, where anomalies in one dimension might be overshadowed by dominant patterns in other dimensions or diluted during the reconstruction process. Such an effect can result in missing anomalies that deviate only in a subset of dimensions. This is illustrated in Figure 1.a. Here, the original signal (indicated by the solid line) contains two anomalies: a subtle anomaly at time point $t_0$ and a significant anomaly at time point $t_1$. The reconstructed signal is slightly smoothed, and by using the reconstruction error alone, the subtle anomaly stays below the detection threshold (see Figure 1.b).

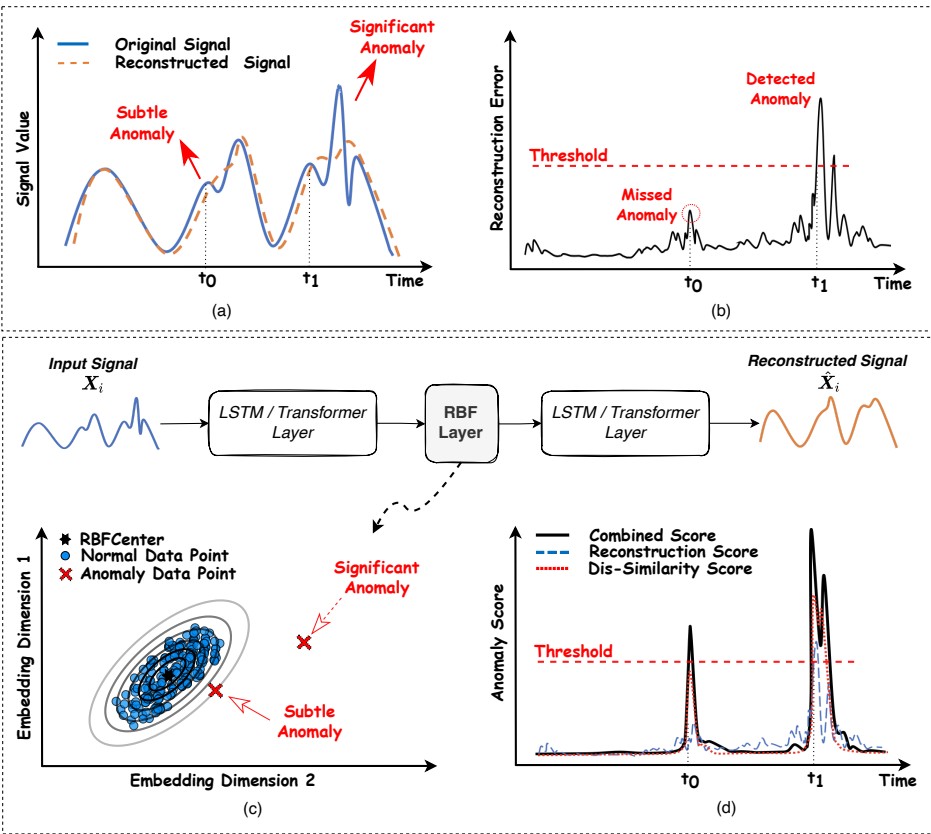

Figure 1: *Challenges with traditional reconstruction-based methods and the proposed RBF kernel solution.*(a) Original time series data with both subtle and significant anomalies, showing the averaging effect in reconstructed signal. (b) Reconstruction error plot using traditional methods, highlighting the challenges in detecting subtle anomalies due to the averaging effect. (c) Model illustration with the RBF kernel integration. The 2D scatter plot displays normal data, subtle and significant anomalies, as well as the RBF center with its influence radius. This highlights RBF kernel's ability to detect deviations from the 'normal' patterns. (d) Anomaly score plot using the combined anomaly score, showcasing enhanced detection capabilities, particularly for subtle anomalies.

Efforts have been made to enhance the efficacy of unsupervised anomaly detection by adding property scores to the conventional reconstruction error anomaly score. For instance, the Anomaly Transformer Xu et al. (2021) introduced the concept of Association Discrepancy, emphasizing that each time point in a series can be described by its associations with all other time points. This innovative approach combines the association discrepancy and reconstruction error to enhance anomaly detection. Notably, while the improved performance of the Anomaly Transformer indirectly underscores the limitations of relying solely on the reconstruction error, their method is specifically tailored for the Transformer architecture and might not serve as a general solution for diverse architectures.

To extend the applicability of an Association Descrepancy-type of anomaly detection to other deep learning methods and deal with the challenges of using RE as the anomaly score, there might still be untapped potential in exploring specialized non-linear transformations, such as the Radial Basis Function (RBF) kernel Orr et al. (1996), for more effective anomaly detection in time series data. The RBF kernel generates a similarity score that measures how close a given data point is to a reference point or center Vert et al. (2004). This can be particularly advantageous for anomaly detection, where anomalies can be intuitively thought of as data points that deviate (are far away) from 'normal' patterns. In this context, anomalies would result in lower similarity scores when evaluated using the RBF kernel, offering a direct measure of deviation. We believe that the RBF similarity computed within a hidden representation of a deep network could offer increased sensitivity to subtle anomalies that might be missed by reconstruction error alone. This is shown in Figure 1.c, where an RBF kernel is fitted on the normal data in the hidden representation. A well-optimized representation allows for the detection of both the subtle and the significant anomalies. By combining the traditional reconstruction error with the RBF similarity score, we create a comprehensive anomaly metric that not only captures deviations from expected patterns but also ensures that subtle anomalies, which might be smoothed out or averaged during reconstruction, are still flagged based on their deviation from the norm. This combined anomaly score is shown in Figure 1.d. The anomaly scores for both anomalies are now above the detection threshold.

In this paper, we focus on incorporating the RBF kernel in LSTM and Transformers baseline models, chosen for their proven efficacy in modeling sequential data, handling temporal dependencies, and widely used in the field. Our goal is to create a new unique anomaly score, which we named *SimRec score*, using both the RBF Similarity score and Reconstruction error. This new score offers a more robust and comprehensive metric for anomaly detection. In summary, this paper offers the following contributions:

- We present *SimRec score*, a composite anomaly score that effectively addresses the shortcomings of solely relying on the reconstruction error by incorporating an RBF similarity score.

- While we focus on LSTM and Transformer models, the design of *SimRec score* is generic and can be applied to a wide range of deep learning architectures.

- We have evaluated our method on various benchmark datasets and shown the superiority of *SimRec score* in detecting anomalies.

## 2 METHODOLOGY

Assume that the observed time series dataset consists of $N$ parts (samples) with the length $T$. Each part of this time series is denoted by $\mathcal{X}_i = \{\boldsymbol{x}_{i,t}\}_{t=1}^T$ where $\boldsymbol{x}_{i,t}$ represents the observed time point for $i$-th sample at time $t$, having $d$ dimensions, i.e., $\boldsymbol{x}_{i,t} \in \mathbb{R}^d$. Our task is to determine if a given $\boldsymbol{x}_{i,t}$ shows any anomalous behavior or not.

### 2.1 RADIAL BASIS FUNCTION (RBF) KERNEL

In the context of our study, the RBF mechanism is integrated into the deep learning framework through a specific layer of RBF neurons. This layer is designed to operate on the hidden representations generated by the preceding layer. The primary role of this single RBF layer is to compute a measure of similarity of a data point with a set of centers (reference points in the learned hidden representation). This similarity is captured using the RBF kernel, which is mathematically represented as:

$$\mathcal{RBF}(\boldsymbol{h}, \boldsymbol{c}) = \exp\left(-\frac{1}{2}e^\gamma \|\boldsymbol{h} - \boldsymbol{c}\|^2\right) \tag{1}$$

Where $\boldsymbol{h}$ represents the hidden representation derived from the preceding layer and $\boldsymbol{c}$ denotes the center of the RBF, which is a learnable parameter. The parameter $\gamma$ is the logarithm of the inverse scale parameter (or the precision parameter), which is initialized and adapted during the training process. The exponential transformation of $\gamma$ in the RBF computation ensures the positivity of the precision parameter for any value for $\gamma$. This stabilizes the optimization of the precision because no positivity constraint has to be enforced. The detailed pseudocode of the RBF Layer is provided in Appendix B.1.

By processing the latent representations by the RBF layer, the model can reason about the similarity of encoded data points with respect to the learned center(s). The final output provides a measure of this similarity. A lower similarity measure (or higher distance) between an encoded representation and the RBF centers could indicate potential anomalies, whereas a higher similarity suggests normal behavior.

### 2.1.1 INITIALIZATION OF RBF PARAMETERS

Initializing the RBF layer parameters, the centers $\boldsymbol{c}$ and to a lesser degree $\gamma$, plays a critical role in our methodology. Initially, we employ a *Random* initialization method, where $\boldsymbol{c}$ and $\gamma$ are initialized using a normal distribution with zero mean and unit standard deviation. While this method is intuitive and straightforward, it might introduce challenges such as slow convergence, the potential risk of local minima, and poor representation of the data distribution during early training, potentially leading to instability.

In addition to the aforementioned approach, we also explored the *K-means* based initialization method. This method uses the inherent structure in the data, if present, to determine initial centers, which can provide a more informed and representative starting point, potentially mitigating some challenges associated with random initialization. In this approach, initially, we focus on training a base model (without the integrated RBF layer) for reconstruction. The objective during this phase is to minimize the Mean Squared Error (MSE) given by:

$$\mathcal{MSE} = \frac{1}{N}\sum_{i=1}^{N}\left\|\mathcal{X}_i - \hat{\mathcal{X}}_i\right\|_F^2 \tag{2}$$

where $N$ is the total number of samples, $\mathcal{X}_i$ represents the actual data, and $\hat{\mathcal{X}}_i$ denotes the model's reconstructed output. The notation $\|\cdot\|_F^2$ denotes the squared Frobenius norm. Once the base model is trained, we extract the hidden representation from the specific layer where we intend to add the RBF layer subsequently in the next phase. This extracted representation is then used for initializing the RBF layer parameters $\gamma$ and $\boldsymbol{c}$. The centers are initialized using the K-means clustering algorithm applied to the extracted hidden representation. The number of clusters is equal to the number of RBF layer centers, denoted by $M$. To initialize $\gamma$, we first compute $\tilde{\sigma}^2$, which represents the average squared distance of each data point in the hidden representation to its nearest center as:

$$\tilde{\sigma}^2 = \frac{1}{NT}\sum_{i=1}^{N}\sum_{t=1}^{T}\min_{j}\|\boldsymbol{h}_{i,t} - \boldsymbol{c}_j\|^2, \quad \forall j \in [1, M] \tag{3}$$

where $\boldsymbol{h}_{i,t}$ denotes the hidden representation vector of the $i$-th sample at the $t$-th time step, and $\boldsymbol{c}_j$ denotes the $j$-th cluster center obtained from the K-means algorithm. This value, $\tilde{\sigma}^2$, is used to initialize $\gamma$ as $\gamma = \frac{1}{\tilde{\sigma}^2}$, ensuring that the RBF function has a spread informed by the average dispersion of the data points around their respective centers.

### 2.2 LEARNING PROCESS

During the training phase of the RBF-integrated model, our primary objective is to ensure accurate reconstruction of the data while also maximizing the likelihood under the RBF kernels. To achieve

this, we minimize a composite loss function that combines MSE and an additional term referred to as the "density loss". The total loss function $\mathcal{L}_{Total}$ is defined as follows:

$$\mathcal{L}_{\text{Total}} = \mathcal{MSE} - \lambda \frac{1}{N} \sum_{i=1}^{N} \frac{1}{T} \sum_{t=1}^{T} \log \left( \frac{1}{M} \sum_{m=1}^{M} (\mathbf{z}_{i,t})_m + \epsilon \right) \tag{4}$$

Here, for each data point $\boldsymbol{x}_{i,t}$, the RBF layer outputs $\mathbf{z}_{i,t} \in \mathbb{R}^M$, where each element $(\mathbf{z}_{i,t})_m$ corresponds to the RBF response of the $m$-th center. The parameter $\lambda$ is a weight factor that controls the importance of the density loss term relative to the overall loss. The term $\epsilon$ is a small constant added to ensure numerical stability during the computation of the logarithm. The logarithmic component of the density loss ensures a balanced RBF response across centers for each data point, preventing any single center from overly dominating the output. This design ensures all data points achieve a high RBF score, enhancing anomaly detection capabilities. This mechanism inherently nudges the centers to position themselves in a way that they effectively 'cover' all data points, ensuring the training data achieves a high likelihood. The intention behind this is twofold: firstly, to ensure that no data point is left inadequately represented, and secondly, to have a high likelihood on the training data, such that anomalies are likely to have a lower likelihood. As a result, the model remains finely tuned to a wide variety of patterns in the data, enhancing its anomaly detection capabilities.

## 2.3 SimRec Anomaly Score

In order to enhance anomaly detection, we incorporate the RBF score and the reconstruction criterion. The RBF score is a measure of similarity between the input data and the learned centers, which can be interpreted as a score indicating the normality of the data. As mentioned, given a sample $\mathcal{X}_i$ the RBF layer output for each data point $\boldsymbol{x}_{i,t}$ is represented by $\mathbf{z}_{i,t} \in \mathbb{R}^M$. The final RBF score measures how close the data point $\boldsymbol{x}_{i,t}$ is to the learned centers:

$$\mathcal{RBF}_{\text{Score}}(\boldsymbol{x}_{i,t}) = \frac{1}{M} \sum_{m=1}^{M} (\mathbf{z}_{i,t})_m \tag{5}$$

A higher RBF score indicates that the data point is more "normal", while a lower score suggests potential anomalies. The reconstruction error for each data point $\boldsymbol{x}_{i,t}$ is given by the squared difference between the actual data and its reconstruction:

$$\mathcal{E}(\boldsymbol{x}_{i,t}) = \| \boldsymbol{x}_{i,t} - \hat{\boldsymbol{x}_{i,t}} \|^2 \tag{6}$$

We compute the RBF score and the reconstruction error for all data points and then normalize them using MinMax normalization to ensure they are on a comparable scale. The final anomaly score, denoted as SimRec, is formulated as:

$$\text{SimRec}(\boldsymbol{x}_{i,t}) = \tilde{\mathcal{E}}(\boldsymbol{x}_{i,t}) \times \left( 1 - \tilde{\mathcal{RBF}}_{\text{Score}}(\boldsymbol{x}_{i,t}) \right) \tag{7}$$

Where $\tilde{\mathcal{E}}(\boldsymbol{x}_{i,t})$ represents the normalized reconstruction error, and $\tilde{\mathcal{RBF}}_{\text{Score}}(\boldsymbol{x}_{i,t})$ is the normalized RBF score. This strategic ombination ensures that subtle anomalies, where both the reconstruction error and RBF score are low, are highlighted, in addition to the significant anomalies with high reconstruction errors or low RBF scores.

## 3 Experimental Setup

### 3.1 Datasets

We use three public benchmark datasets for our experiments. 1) *Server Machine Dataset (SMD)* Su et al. (2019): Collected from an Internet company, comprising data from 38 sensors across 28 server machines. The training and test sets are of equal size. Labels are provided to indicate anomalous points, and every dimension contributes to the anomaly parts. 2) *Mars Science Laboratory (MSL)*

*Rover* Hundman et al. (2018): An expert-labeled dataset of 55 dimensions, collected from NASA's Incident Surprise Anomaly (ISA) reports. 3) *Pooled Server Metrics (PSM)* Abdulaal et al. (2021): Collected internally from multiple application server nodes at eBay. The data consists of 25 features representing server machine metrics, such as CPU usage and memory. The anomaly labels are manually labeled by experts. For detailed statistics on each dataset, please refer to Appendix A.

## 3.2 Data Preprocessing

Initially, each signal in the dataset is normalized to zero mean and unit variance, performed across the time dimension. Subsequently, following the protocol in Shen et al. (2020), the normalized signal is segmented into non-overlapped sliding windows with a fixed length of 100 data points (a common setting based on the previous related works).

## 3.3 Implementation

### 3.3.1 Models

We employ LSTM and Transformer models as our baseline architectures inspired by the research works of Hundman et al. (2018) and Xu et al. (2021). The LSTM model comprises two LSTM layers, with a dropout layer following the first LSTM layer, and a fully connected layer as the final layer. Each LSTM layer has 80 hidden states. Building upon this architecture, we introduce the SimRec-LSTM by incorporating RBF layer between the two LSTM layers, positioned right after the dropout layer. For our Transformer model, we begin with a DataEmbedding module that combines both token and positional embeddings. This is followed by an encoder made up of three layers. Each of these layers is equipped with a multi-head self-attention mechanism and feed-forward networks. Specifically, the model's hidden state dimension is set to 32, and the intermediate layer of the feed-forward networks within the Transformer blocks has a dimension of 128. The number of attention heads is set to 8. In our proposed SimRec-Transformer model, we incorporate an RBF layer between the second and third encoder layers. For a detailed visual representation of these models, please refer to Appendix B.2.

We use the ADAM optimizer for optimization. We determined the hyperparameters for all models by systematically searching through all possible combinations to achieve optimal performance on the reconstruction task. Further details regarding the hyperparameters can be found in Appendix B.3.

### 3.3.2 Evaluation

We label time points as anomalies if their anomaly score, as defined in Equation 7, exceeds a threshold, $\delta$. Following the approach used by Xu et al. (2021), we set $\delta$ to label a predefined proportion $r$ of the validation dataset as anomalies. This approach is practical and efficient in real-world applications where the number of anomalies that can be investigated is often decided by human resources. Given that the aforementioned study is a state-of-the-art reference, we adopt their suggested ratios for our datasets. Specifically, we set $r$ to 0.5% for the SMD dataset, and 1% for other datasets.

We employ the widely-adopted "point-adjust" approach for evaluating anomaly detection in time series, as introduced by Xu et al. (2018) and used widely in subsequent works like Su et al. (2019); Xu et al. (2021). This method acknowledges the contiguous nature of anomalies in time series data. We evaluate the performance using the F1-score metric and further employ Area Under the Receiver Operating Characteristic Curve (AUC-ROC), Area Under the Precision-Recall Curve (AUC-PR) for a threshold-independent evaluation. These metrics offer a broader perspective and reduce dependency on specific threshold settings. Additionally, we have adopted innovative metrics such as Volume Under the Surface of the Receiver Operating Characteristic Curve (VUS-ROC) and Volume Under the Surface of the Precision-Recall Curve (VUS-PR) Paparrizos et al. (2022). These measures are particularly designed for time-series anomaly detection, providing a robust, parameter-free evaluation that is not influenced by threshold choices.

## 4 Results

Our empirical findings, presented in Table 1, demonstrate the benefits of integrating the RBF layer into baseline models and employing the SimRec score for anomaly detection in time series data.

Both the SimRec-LSTM and SimRec-Transformer models, whether initialized randomly or using K-means clustering, consistently outperform their respective baselines across all benchmark datasets. The relatively small standard deviations across multiple runs indicate that the performance improvements are stable.

Table 1: *Mean performance metrics (in %) of baseline and RBF-integrated models on test sets over five training runs - Mean(std).* Initialization methods are denoted as (R) for Random and (K) for K-means. A higher value in performance metrics indicates better performance.

| Dataset | SMD | | | | | MSL | | | | | PSM | | | | |
|---|---|---|---|---|---|---|---|---|---|---|---|---|---|---|---|
| Models | F1-Score | AUC | AUC-PR | VUS-ROC | VUS-PR | F1-Score | AUC | AUC-PR | VUS-ROC | VUS-PR | F1-Score | AUC | AUC-PR | VUS-ROC | VUS-PR |
| LSTM | 76(1) | 85(0) | 59(1) | 61(0) | 55(0) | 82(3) | 87(2) | 70(4) | 72(1) | 71(1) | 91(3) | 92(2) | 88(3) | 67(2) | 76(1) |
| SimRec-LSTM (R) | 80(3) | 88(2) | 65(4) | 65(2) | 60(3) | 88(1) | 92(1) | 80(1) | 78(1) | 78(1) | 95(4) | 96(4) | 93(5) | 84(2) | 88(1) |
| SimRec-LSTM (K) | 82(3) | 89(2) | 68(5) | 66(2) | 61(2) | 89(1) | 92(1) | 80(1) | 80(0) | 79(1) | 96(1) | 96(1) | 94(2) | 87(1) | 89(0) |
| Transformer | 73(3) | 83(2) | 56(5) | 59(2) | 52(3) | 80(3) | 86(2) | 68(4) | 70(2) | 70(2) | 84(4) | 87(3) | 80(4) | 65(2) | 75(1) |
| SimRec-Transformer (R) | 77(4) | 87(3) | 61(5) | 64(3) | 57(3) | 88(0) | 92(0) | 80(1) | 80(1) | 79(1) | 91(1) | 92(1) | 87(1) | 71(3) | 79(2) |
| SimRec-Transformer (K) | 82(3) | 88(3) | 69(4) | 66(2) | 61(4) | 88(1) | 92(1) | 79(2) | 79(2) | 78(2) | 90(5) | 91(4) | 86(5) | 69(5) | 79(4) |

Figure 2 visually demonstrates the advantage of the SimRec score, emphasizing its enhanced sensitivity and precision compared with the traditional reconstruction error. The SimRec score identifies the anomalies that are missed or overlooked by the baseline methods using only the reconstruction error. These results underscore the efficacy of the RBF kernel in distinguishing deviations that might be challenging to detect based solely on reconstruction error.

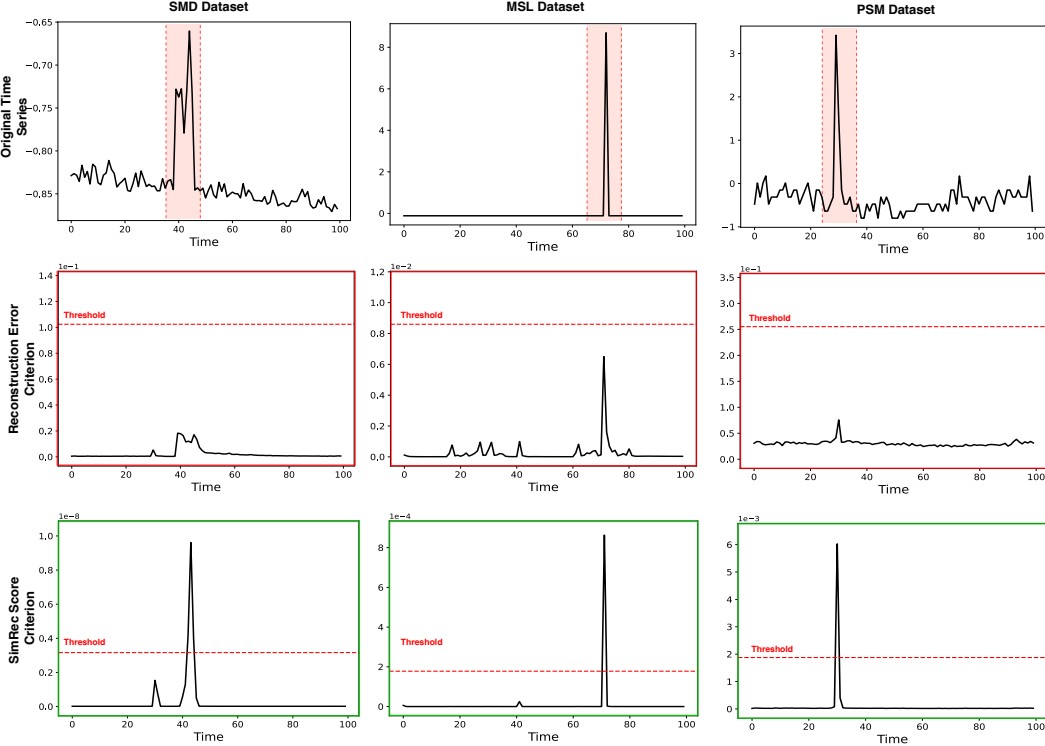

Figure 2: *Comparison of the reconstruction error and SimRec score in anomaly detection across all datasets.* The figure depicts instances where the SimRec score successfully detects anomalies that are missed by the baseline reconstruction error method. The figure showcases just one example among several from each of the datasets. Note that the original signal represents only a single feature selected from multiple available features. Anomalous sections in the original signal are highlighted in red. The plots are scaled for visualization clarity.

## 4.1 ABLATION ANALYSIS

In an ablation study, we investigated the influence of $\lambda$, which represents the weight of the density loss term in our loss function (see equation 4). Figure 3 demonstrates the performance of the SimRec-LSTM model with both random and K-means initialization across various $\lambda$ values.

Across most of the datasets and both initialization methods, we observed a trend where the performance generally increases as $\lambda$ increases. This trend suggests that giving more weight to the density loss is beneficial for the anomaly detection task in most scenarios. However, a notable exception was observed with the MSL dataset when the K-means initialization method was employed. In this case, as $\lambda$ increases, the performance metrics show a slight decrease. This behavior underscores the importance of dataset-specific considerations and indicates that while increasing $\lambda$ might be advantageous in many situations, it might not universally hold true. Additionally, when comparing initialization strategies, K-means initialization shows a relatively more stable performance across different $\lambda$ values compared to random initialization.

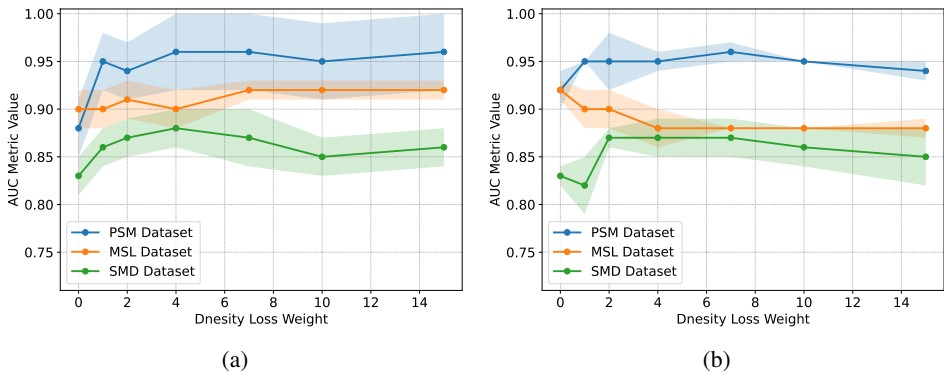

(a)                                          (b)

Figure 3: *Mean AUC performance of SimRec-LSTM over different $\lambda$ values for both initialization strategies over five different training runs.* (a) Random Initialization (b) K-means Initialization.

In addition, we explored the flexibility of the RBF layer placement within the baseline architectures. For the LSTM, we initially placed the RBF layer after the first LSTM layer ('Middle-Placement'). In the ablation setup, we placed the RBF layer after the second LSTM layer, preceding the fully-connected layer ('End-Placement'). For the Transformer, 'Middle-Placement' denotes the RBF layer after the second encoder layer, whereas 'End-Placement' is after the third encoder layer, before the final fully-connected layer. The results are presented in Figure 4. Across all datasets, both models consistently maintained their performance, irrespective of the location of the RBF layer. These findings highlight that the SimRec score retains its effectiveness in anomaly detection regardless of the position of the RBF layer within the model. (See Appendix C for more ablation analyses.)

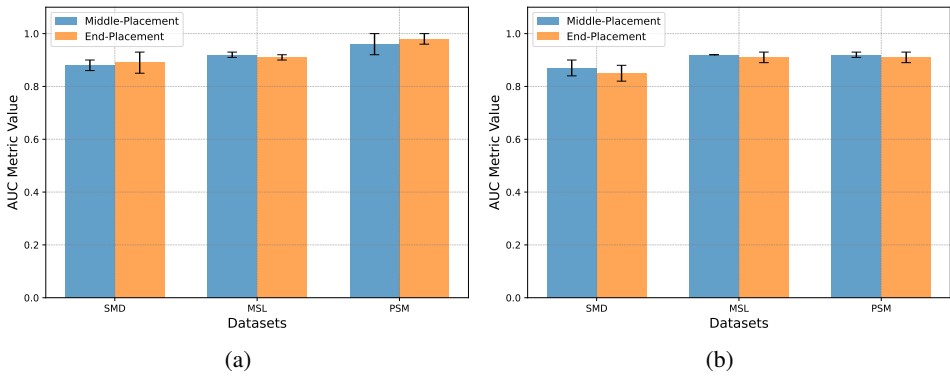

(a)                                          (b)

Figure 4: *Mean AUC performance of SimRec-LSTM and SimRec-Transformer models with 'Middle-Placement' and 'End-Placement' configurations of the RBF layer over five different training runs.* (a) SimRec-LSTM (b) SimRec-Transformer.

## 5 COMPARISON WITH STATE-OF-THE-ART

Although the primary objective was to address the foundational issue of the smoothing effect in reconstruction-based anomaly detection, we also highlighted the comparative performance. The performance of our proposed method was evaluated against a recent state-of-the-art model; the Anomaly Transformer Xu et al. (2021), as shown in Table 2[1]. Although our method shows a lower performance on the SMD dataset, its performance on the MSL and PSM dataset is very competitive. Notably, we used simple baseline architectures without any task-specific optimization, unlike many state-of-the-art models. Given this, the competitive performance of our method indicates its potential strength. This suggests that with tailored architectures for the task, our approach could potentially bridge the gap or even surpass current benchmarks.

Table 2: *Comparison of SimRec method with the state-of-the-art Anomaly Transformer - Mean(std).* The table presents the mean performance metrics (in %) over five different training runs.

| Dataset | SMD | | | | | MSL | | | | | PSM | | | | |
|---|---|---|---|---|---|---|---|---|---|---|---|---|---|---|---|
| Models | F1-Score | AUC | AUC-PR | VUS-ROC | VUS-PR | F1-Score | AUC | AUC-PR | VUS-ROC | VUS-PR | F1-Score | AUC | AUC-PR | VUS-ROC | VUS-PR |
| Anomaly Transformer | 88(1) | 93(1) | 79(1) | 76(1) | 72(1) | 90(1) | 94(1) | 83(2) | 82(2) | 81(2) | 96(1) | 96(1) | 93(1) | 82(1) | 86(1) |
| SimRec-LSTM | 82(3) | 89(2) | 68(5) | 66(2) | 61(2) | 89(1) | 92(1) | 80(1) | 80(0) | 79(1) | 96(1) | 96(1) | 94(2) | 87(1) | 89(0) |

## 6 DISCUSSION AND CONCLUSION

In this study, we introduced the SimRec score that combines a similarity score and reconstruction error to detect anomalies in time series data, addressing the limitation of reconstruction error anomaly score. The similarity score is obtained by a nonparametric density estimate using a layer of RBF neurons in a deep network. This anomaly score increases the sensitivity to subtle anomalies that are missed by the reconstruction error. The location of this RBF layer in the network is not sensitive, and this generic setup makes it possible to apply the SimRec score to any arbitrary deep network.

In an ablation analysis, we showed that the initialization of the RBF kernel is important. The K-means initialization of the RBF layer generally showed higher stability (and often better performance) than random initialization. However, due to the complexity of the K-means implementation, random initialization can be a good alternative.

To optimally use this SimRec score, two measures have to be optimized: the reconstruction error and the similarity score. The trade-off between these two measures is, unfortunately, dependent on dataset characteristics and RBF layer initialization. In two datasets (PSM and SMD), a higher weight for the similarity loss is required, while for the MSL dataset, the reconstruction error has a higher priority. This can be attributed to characteristics of the MSL dataset, such as high dimensionality and few number of training samples, which might have led to less effective initialization using the K-means method.

The SimRec score, in comparison with the state-of-the-art method showed competitive performance, especially on MSL and PSM datasets. This competitive performance, achieved with not-tailored architectures for the task, highlights the effectiveness and potential for enhancement of the proposed method. Future research can focus on developing architectures specifically tailored for the SimRec score to potentially unlock superior performance.

To conclude, this research addresses a fundamental challenge in time series reconstruction-based anomaly detection by introducing the SimRec score- a simple, flexible, yet profoundly effective metric. Our findings not only highlight the clear performance improvement over the baseline models but also lay the groundwork for future research. The current approach combines the reconstruction error and similarity score using a multiplicative method. Future studies could delve into alternative methods of combination, such as additive, weighted average, or even more complex fusion strategies. Moreover, extending the evaluation to a broader range of datasets and baselines will be a valuable direction for future research.

---

[1]We executed the Anomaly Transformer with details described in the published paper, and averaged results over five runs under our settings for fairness, due to the lack of multi-run results in the original study.

## REPRODUCIBILITY STATEMENT

Dedicated to ensuring the reproducibility of our research, we have taken several careful steps. We provided the source code that offers detailed implementations of the baseline and RBF-integrated model architectures, ensuring complete transparency in our methodologies and experiments. This code also encompasses comprehensive details on training configurations and hyperparameters. Additionally, the data processing steps are thoroughly outlined within the code itself. For further clarity on the hyperparameters and training specifics, they are also elaborated upon in the Appendix section of the paper. Delving into the theoretical aspect, the main text explains the mathematical derivations and foundations behind our work. In our experimentation, we have made sure to average results over multiple runs to guarantee the fairness and reproducibility of our findings, eliminating any dependence on specific random seed configurations. All datasets used in our study are publicly available. By offering these details, we aim to promote transparency and facilitate further exploration by the research community. Should any questions arise, we encourage researchers to contact the corresponding author for further clarification.

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

# A   DATASETS

In Table 3, we provide detailed statistics for each of the datasets used in our experiments. This includes the number of dimensions, the size of the training and test datasets, and the proportion of actual anomalies (Anomaly Rate).

Table 3: *Statistics and characteristics of benchmark datasets used in experiments.* Note: # indicates the number of windows after the windowing process. The Anomaly Rate is based on the original data points.

| Benchmarks | Applications | Dimension | #Training | #Test Samples | Anomaly Rate |
|:---:|:---:|:---:|:---:|:---:|:---:|
| SMD | Server | 38 | 7084 | 7084 | 0.042 |
| MSL | Space | 55 | 583 | 737 | 0.105 |
| PSM | Server | 25 | 1324 | 878 | 0.278 |

# B   IMPLEMENTATION DETAILS

## B.1   RBF LAYER PSEUDOCODE

We present a detailed pseudocode of the RBF layer in Algorithm 1. This representation elucidates the step-by-step operations for computing the RBF kernel values, based on the input data and the set of RBF centers.

---
**Algorithm 1** RBF Layer Forward Pass

---
**Require:**
   $\mathcal{X} \in \mathbb{R}^{T \times d}$: Input matrix where $T$ is the number of data points and $d$ is the dimensionality of each data point.
   $\mathcal{C} \in \mathbb{R}^{m \times d}$: Matrix of RBF centers where $m$ is the number of centers.
   $\log(\gamma)$: Logarithm of the inverse scale parameter.
**Ensure:**
   $\mathcal{O} \in \mathbb{R}^{T \times m}$: Output matrix of RBF kernel values.

1: Compute squared Euclidean distance matrix $\mathcal{D}$ between rows of $\mathcal{X}$ and $\mathcal{C}$ as:

$$\mathcal{D}_{ij} = \|\mathcal{X}_{i,:} - \mathcal{C}_{j,:}\|^2, \quad \forall i \in [1, T], \forall j \in [1, m]$$

$\triangleright \mathcal{D} \in \mathbb{R}^{T \times m}$

2: Compute RBF kernel matrix $\mathcal{O}$ as:

$$\mathcal{O} = \exp\left(-0.5 \cdot \exp(\log(\gamma)) \cdot \mathcal{D}\right)$$

$\triangleright \mathcal{O} \in \mathbb{R}^{T \times m}$

3: **return** $\mathcal{O}$                         $\triangleright$ Represent the similarity scores as output of the RBF layer

---

## B.2   MODEL ARCHITECTURES

Figure 5 provides a visual representation of the architectures of both the SimRec-LSTM and SimRec-Transformer models.

## B.3   HYPERPARAMETERS

We determined the hyperparameters for both the proposed models (SimRec-LSTM and SimRec-Transformer) and the baseline models through a systematic grid search to ensure optimal reconstruction performance. The batch size is set to 128 for all models. For the proposed SimRec models, training is conducted for 100 epochs, while the baseline models are trained for 500 epochs with an early stopping criterion of 100 epochs to prevent overfitting. In all training settings, we employed

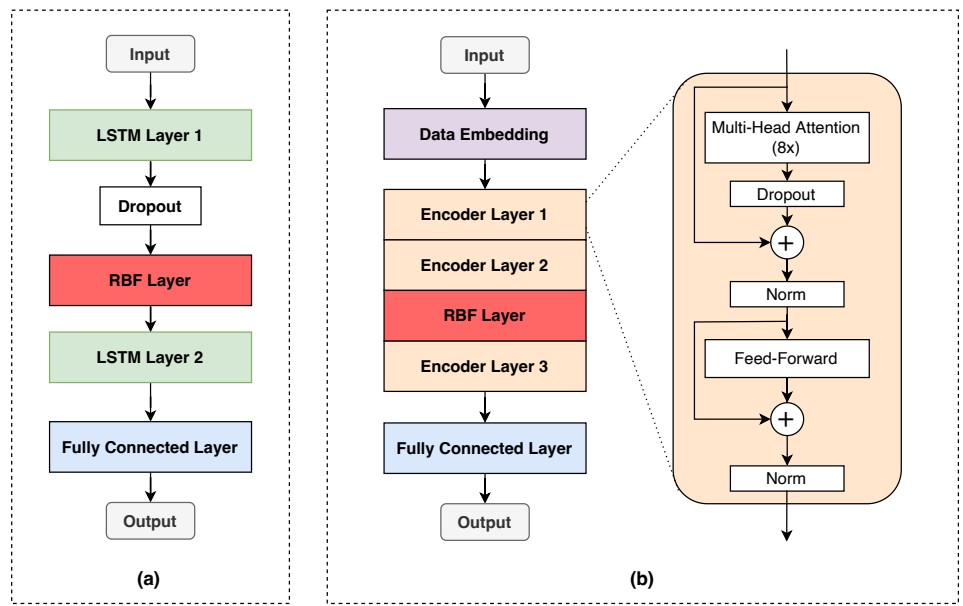

Figure 5: *Visual representations of the (a) SimRec-LSTM and (b) SimRec-Transformer models employed in our experiments.* The detailed view of the internal components of a single Transformer encoder layer is also depicted, shown on the right side of the Transformer model diagram.

a learning rate scheduler that reduces the learning rate by a factor of 0.5 if there is no improvement in validation performance after 10 epochs. The specific hyperparameters tailored to each dataset for the baselines and proposed SimRec models are presented in Table 4.

Table 4: *Hyperparameters for Baseline and SimRec LSTM and Transformer across datasets using both random and K-means initializations.*

| Dataset | Model | lr | weight decay | dropout | clip grad | $\lambda_{Random}$ | $\lambda_{Kmeans}$ | RBF centers |
|---|---|---|---|---|---|---|---|---|
| SMD | Baseline-LSTM | $10^{-3}$ | $10^{-5}$ | 0.3 | 1 | - | - | - |
| | Baseline-Transformer | $10^{-3}$ | $10^{-3}$ | 0 | 1 | - | - | - |
| | SimRec-LSTM | $10^{-1}$ | $10^{-4}$ | 0.3 | 3 | 4 | 5 | 128 |
| | SimRec-Transformer | $10^{-3}$ | $10^{-5}$ | 0 | 1 | 5 | 10 | 512 |
| MSL | Baseline-LSTM | $10^{-2}$ | $10^{-5}$ | 0 | 1 | - | - | - |
| | Baseline-Transformer | $10^{-3}$ | $10^{-2}$ | 0.3 | 1 | - | - | - |
| | SimRec-LSTM | $10^{-2}$ | $10^{-5}$ | 0 | 0.5 | 7 | 0 | 32 |
| | SimRec-Transformer | $10^{-2}$ | $10^{-2}$ | 0 | 0.5 | 0 | 6 | 16 |
| PSM | Baseline-LSTM | $10^{-3}$ | $10^{-4}$ | 0 | 1 | - | - | - |
| | Baseline-Transformer | $10^{-3}$ | $10^{-6}$ | 0.3 | 1 | - | - | - |
| | SimRec-LSTM | $10^{-2}$ | $10^{-1}$ | 0 | 1.5 | 7 | 7 | 16 |
| | SimRec-Transformer | $10^{-2}$ | $10^{-1}$ | 0 | 1.5 | 1 | 0 | 64 |

# C    ABLATION STUDY: VARYING NUMBER OF RBF CENTERS

In a further ablation study, we investigated the performance of the SimRec-LSTM by changing the number of centers in the RBF layer, shown in Figure 6. Results show that while the optimal number of RBF centers varies across datasets, it appears that there's a threshold beyond which adding more centers does not result in performance improvements and might even degrade the performance. The optimal number of RBF centers is influenced by the initialization method and appears to show data-specific dependencies. In general, K-means initialization performance is more stable compared to random initialization, especially with a small number of RBF centers.

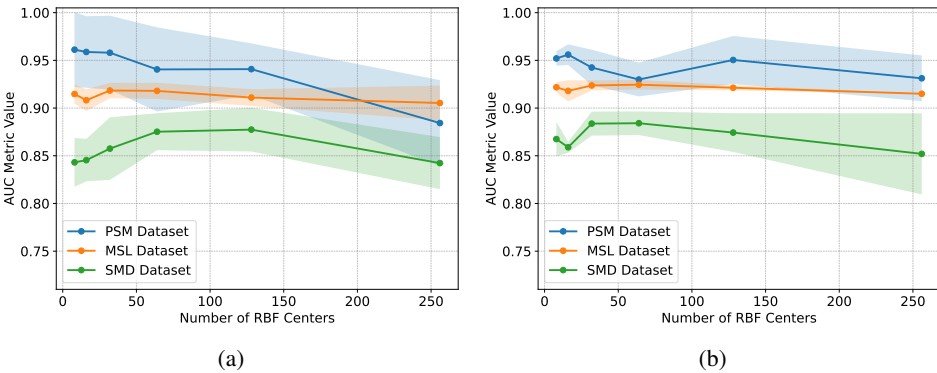

Figure 6: *Performance of SimRec-LSTM over different numbers of RBF centers for both initialization strategies.* (a) Random Initialization (b) K-means Initialization.

