# OpenReview forum: "Time Series Anomaly Detection using Reconstruction and RBF Similarity Scores"
_ICLR.cc/2024/Conference — Submitted to ICLR 2024_

### Official Review · Reviewer_ewvW · 2023-10-23

**Soundness:** 2 fair
**Presentation:** 3 good
**Contribution:** 2 fair
**Rating:** 5
**Confidence:** 3

**Summary:**

In order to mitigate the impact of smoothing effects and improve the network's sensitivity to anomalies, the author brings RBF to current time-series anomaly detection methods like LSTM and Transformer. By constructing a clustering-based middle layer with RBF, all the input samples are projected to cluster space for comparing their distances to cluster centroids, which is benefit for separating anomalies from samples. Besides, an RBF-similarity-score-based criterion is proposed for optimizing the parameters as well as evaluating the effect of proposed methods.

**Strengths:**

Strengths:
1)	The motivation of the work is quite direct and practical in anomaly detection fields, which is easy to catch up;
2)	Experiments with LSTM and Transformer on 3 different datasets demonstrate the effectiveness of proposed methods, and the performance improvement brought by the proposed module is stable;
3)	Ablation study on initialization methods, inserting position and number of cluster centroids firmly analyzed the effectiveness of proposed methods under different circumstances;

**Weaknesses:**

1)	In review part, only Xu et al. (2021) is reviewed for describing current time-series anomaly detection methods, but several works in 2022 and 2023 are missed in reviewing and experimental parts, which have also performed greatly in multiple time-series anomaly detection datasets, such as:
[1] Zhang Z, Li W, Ding W, et al. STAD-GAN: unsupervised anomaly detection on multivariate time series with self-training generative adversarial networks[J]. ACM Transactions on Knowledge Discovery from Data, 2023, 17(5): 1-18.
[2] Xia F, Chen X, Yu S, et al. Coupled Attention Networks for Multivariate Time Series Anomaly Detection[J]. IEEE Transactions on Emerging Topics in Computing, 2023.
[3] Ding C, Sun S, Zhao J. MST-GAT: A multimodal spatial–temporal graph attention network for time series anomaly detection[J]. Information Fusion, 2023, 89: 527-536.
2)	The experiments were only conducted on LSTM and Transformer, which are some out of-date baselines, more experiments on current SOTA methods (as baselines) should be conducted for validating the performance of proposed module, since the author mentioned that the proposed method is generic and can be applied to a wide range of deep learning architectures. For instance, the Anomaly Transformer can be regarded as a good baseline to testify the proposed RBF-based module.

**Questions:**

Refering to above

---

> ### Author Response · Authors · 2023-11-18
> **Response to Reviewer ewvW**
>
> Thank you for your valuable suggestions and comments regarding our paper. We have carefully considered your feedback and updated our manuscript accordingly.
>
> ### **Q1. Miss of related Works:**
>
> Following your suggestions, we have updated the related research section in the introduction of our paper to include the recent works and baselines. This addition has enriched our paper by providing a more comprehensive overview of the recent methods.
>
> &nbsp;
>
> ### **Q2. Lack of Baseline** (Application of SimRec in other baselines):
>
> We understand your curiosity about the application of our SimRec method to more specialized anomaly detection models. We address your point in the following details:
>
> - **Rationale Behind the Selected Baselines:** Our study specifically focuses on enhancing the anomaly detection in models based on Reconstruction Error (RE). This focus derives from the identified gap where these models often struggle with detecting subtle anomalies. Our selection of LSTM and Transformer baselines is relevant to the scope of the paper and is driven by their widespread use as RE-based benchmarks in anomaly detection literature. Many papers employ these models as reconstruction-error based benchmarks when comparing new methods. This choice allows us to demonstrate the significant improvements our SimRec method can bring to widely recognized models in the field of RE-based anomaly detection.
>
> - **Exploring Beyond Initial Scope (Potential Application to Specialized Models):** In response to your suggestion, we appreciated the potential value of exploring the application of SimRec to more specialized models like Anomaly Transformer or Omny, despite them operating on different principles such as association discrepancy. This exploration was not part of our original research plan but was undertaken to address your feedback.
>   - **Challenges with Implementation:** We encountered challenges while attempting to apply SimRec to the Anomaly Transformer model due to discrepancies between the published paper and the public code. Notably, the model's code includes a "temperature" parameter set to a value of 50, a critical detail that is absent in the original paper. When we adjusted this parameter to a value of 1, more in line with standard practice and implied by the paper, we observed a substantial drop in performance on datasets like MSL. For instance, the F1-score decreased from 0.95 to 0.88, and the AUC from 0.98 to 0.92. This discrepancy highlights the difficulties in replicating and adapting methods when crucial implementation details are omitted in the publication.
>
>     Despite reaching out for clarification, we have not received a response, which makes it hard to properly integrate SimRec with their model. The limited time frame of the rebuttal period and complexities in adapting SimRec due to unclear published core and details constrained our ability to extend our research in this direction. This point brings us back to the choice of our baselines as they were not just arbitrary but a strategic decision to demonstrate the impact of SimRec within a well-recognized and relevant framework in the field.
>
> - **Focused Approach with Openness to Future Research:** We recognize the potential benefits of integrating SimRec with the specialized models, but it should be noted that our research is dedicated to addressing challenges within the pure RE-based anomaly detection domain. We would like to keep the scope of the study in establishing the foundational effectiveness of SimRec in the targeted context of RE score. We believe this focused approach better illustrates the impact of SimRec. Nevertheless, exploring the applicability of SimRec to a broader range of models, including specialized ones, is an exciting direction for future research which can be now opened by our study. We updated the paper accordingly to show this point.
>
> We hope that the revisions and provided responses address your concerns adequately.

---

> > ### Comment · Reviewer_ewvW · 2023-11-20
> > **Comments/suggestions**
> >
> > 1)	In the mentioned rationale the author claimed that LSTM and Transformer are universally adopted as RE-based benchmark, but in recent years GAN and AE based time series anomaly detection methods, which also utilize reconstruction error as one of their principles, have been proposed and become mainstream methods, I wonder if your proposed SimRec can work well with GAN or AE based methods.
> > 2)	The reproduction of Anomaly Transformer is to set another baseline for you to validate the effectiveness of your proposed method with other baselines. No matter how unstable the baseline is, if your method can improve its performance according to your reproduced results (under the same experimental settings), the effectiveness can still be proved. Besides, you can also try to test the performance of your module with different “temperature” settings. If your method can stably improve the performance of the baseline, then the stability and effectiveness can be further testified.
> > 3)	Except Anomaly Transformer, I also find another transformer-based baseline for you to test your method.
> > [1] Tuli S, Casale G, Jennings N R. TranAD: deep transformer networks for anomaly detection in multivariate time series data[J]. Proceedings of the VLDB Endowment, 2022, 15(6): 1201-1214.
> > Arxiv: https://arxiv.org/pdf/2201.07284.pdf
> > Code: https://github.com/imperial-qore/TranAD
> > The methods are experimented on multiple time-series AD datasets (including the ones in your article), which is also a good baseline method for validation.

---

> > > ### Author Response · Authors · 2023-11-22
> > > **New Response to Reviewer ewvW**
> > >
> > > We appreciate your continued engagement and insightful suggestions, especially the **TranAD paper**, as this can be really helpful. We understand the perspective that applying our SimRec method to newer, state-of-the-art baselines could further validate its effectiveness. We would like to address this point comprehensively:
> > >
> > > - **Demonstrating Core Concept on Established Models:**
> > > Our study’s primary objective is to introduce and validate the SimRec concept. To achieve this, we deliberately selected widely-recognized models like LSTM and Transformer. This choice was made to provide a clear baseline for demonstrating SimRec's impact, ensuring that any performance improvements can be directly attributed to our method.
> > >
> > >     - Integrating SimRec with more complex, state-of-the-art models introduces significant technical and conceptual challenges. These complexities could potentially lead to difficulties in isolating the impact of SimRec, complicating the fair and meaningful assessment of its efficacy. Our approach, therefore, focused on ensuring a clear and unambiguous evaluation of SimRec's effectiveness. While exploring SimRec’s applicability to cutting-edge models is an exciting direction, our current focus is on **establishing its foundational effectiveness** in a controlled and comparably simpler environment.
> > >
> > >
> > >     - Regarding the suggested models (e.g., TranAD), it can be seen that they already demonstrate high AUC performance (0.99). We must carefully consider what additional benefits SimRec could bring, maybe improvements in aspects like interpretability, robustness, or efficiency. However, pursuing this path leads us again away from the goal of the paper. Also, integrating SimRec could necessitate altering these models' architectures, which might raise questions about fairness and the direct comparability of results (One might then question if the selected baseline is now different from the proposed tailored and optimized method).
> > >
> > >
> > > - **Future Research Directions:**
> > > We fully acknowledge the importance of testing SimRec with newer models as a crucial step. We believe this direction holds significant promise for us and other researchers in the field to apply SimRec to a broader range of advanced models, including those suggested.
> > >
> > >
> > > In summary, while we recognize the benefits of applying SimRec to new baselines, our current research is focused on establishing its foundational effectiveness.
> > >
> > > Thank you for your understanding and support of our research direction. We hope that our response addresses your concerns and look forward to the possibility of our work contributing to the conferance.

---

### Official Review · Reviewer_L6VE · 2023-10-29

**Soundness:** 3 good
**Presentation:** 3 good
**Contribution:** 2 fair
**Rating:** 5
**Confidence:** 4

**Summary:**

Until recently, anomaly detection was mainly done using unsupervised learning and autoencoder-based methods due to the lack of label data. Learning methods based on reconstruction error cannot detect subtle anomalies in multivariate dataset/high-dimensional dataset.

In this paper, to solve these limitations, Radial Basis Function (RBF) neurons are applied to deep learning architecture.

In other words, this paper proposes a new anomaly score called SimRec. SimRec is an anomaly score that combines RBF score and reconstruction error. By using this score, subtle existing anomalies can be detected.

**Strengths:**

1. In particular, in unsupervised anomaly detection, a motif to overcome the limitations of reconstruction error-based anomaly detection is reasonable.
2. The paper is overall understandable and neatly written.

**Weaknesses:**

1. There is a lack of baseline. Since neither LSTM nor Transformer are models specialized for anomaly detection, I am curious about the results when applied to more specialized models. In particular, I would like to see the results of applying SimRec to the Anomaly Transformer. In addition, Omni Anomaly
2. The effect of SimRec is clearly visible, but it seems to be more affected by threshold selection. I would like to see the reconstruction error graph of the anomaly transformer using the same threshold selection method.
3. This paper clearly has good motivation, but it feels somewhat lacking in terms of experimental performance, experimental content, and contribution.

**Questions:**

1. There is a lack of baseline. Since neither LSTM nor Transformer are models specialized for anomaly detection, I am curious about the results when applied to more specialized models. In particular, I would like to see the results of applying SimRec to the Anomaly Transformer. In addition, Omni Anomaly
2. The effect of SimRec is clearly visible, but it seems to be more affected by threshold selection. I would like to see the reconstruction error graph of the anomaly transformer using the same threshold selection method.

---

> ### Author Response · Authors · 2023-11-18
> **Response to Reviewer L6VE**
>
> Thank you for your insightful feedback. We carefully addressed them point-by-point and tried to make possible revisions accordingly.
>
> ### **Q1. Lack of Baseline** (Application of SimRec in other baselines):
>
> We understand your curiosity about the application of our SimRec method to more specialized anomaly detection models. We address your point in the following details:
>
> - **Rationale Behind the Selected Baselines:** Our study specifically focuses on enhancing the anomaly detection in models based on Reconstruction Error (RE). This focus derives from the identified gap where these models often struggle with detecting subtle anomalies. Our selection of LSTM and Transformer baselines is relevant to the scope of the paper and is driven by their widespread use as RE-based benchmarks in anomaly detection literature. Many papers employ these models as reconstruction-error based benchmarks when comparing new methods. This choice allows us to show the significant improvements our method can bring to widely recognized models in the field of RE-based anomaly detection.
>
> - **Exploring Beyond Initial Scope (Potential Application to Specialized Models):** In response to your suggestion, we appreciated the potential value of exploring the application of SimRec to more specialized models like Anomaly Transformer or Omny, despite them operating on different principles such as association discrepancy. This exploration was not part of our original research plan but was undertaken to address your feedback.
>
>   - **Challenges with Implementation:** We faced challenges while trying to apply SimRec to the Anomaly Transformer model due to discrepancies between the published paper and the public code. Notably, the model's code includes a "temperature" parameter set to a value of 50, a critical detail that is absent in the original paper. When we adjusted this parameter to a value of 1, more in line with standard practice and implied by the paper, we observed a substantial drop in performance on datasets like MSL. For instance, the F1-score decreased from 0.95 to 0.88, or the AUC from 0.98 to 0.92. This discrepancy highlights the difficulties in replicating and adapting methods when crucial implementation details are omitted in the publication.
>
>     Despite reaching out for clarification, we have not received a response, which makes it hard to properly integrate SimRec with their model. The limited time frame of the rebuttal period and complexities in adapting SimRec due to unclear published core and details constrained our ability to extend our research in this direction. This point brings us back to the choice of our baselines as they were not just arbitrary but a strategic decision to show the impact of SimRec within a well-recognized and relevant framework in the field.
>
> - **Focused Approach with Openness to Future Research:** We recognize the potential benefits of integrating SimRec with the specialized models, but it should be noted that our research is dedicated to addressing challenges within the pure RE-based anomaly detection domain. We would like to keep the scope of the study in establishing the foundational effectiveness of SimRec in the targeted context of the RE score. We believe this focused approach better illustrates the impact of SimRec. Nevertheless, exploring the applicability of SimRec to a broader range of models, including specialized ones, is an exciting direction for future research which can be now opened by our study. We updated the paper accordingly to show this point.
>
> &nbsp;
>
> ### **Q2. Effect of the Threshold**
>
> - **Threshold Selection Methodology:** In our research, the threshold for anomaly detection was carefully chosen based on the well-accepted method used in the literature, specifically referencing the approach used in the Anomaly Transformer paper.
>
> - **Use of ROC-AUC and AUC-PR:** To provide a more comprehensive evaluation that is not solely dependent on threshold selection, we have employed ROC-AUC and AUC-PR metrics which offer a threshold-independent assessment and a general view of the effect of our method.
>
> - **Incorporation of New Evaluation Metrics:** Following the suggestion of reviewer \#GeHM, we have added new evaluation metrics to our paper that are not dependent on threshold selection. These additional results align with our initial findings and support the claims of our study. Results can be found in Table 1 in the updated paper.
>
> - **Consistency Across Comparisons:** We have ensured that the comparison of our method with other models is fair by using the same threshold and settings. Also, to provide a robust evaluation, we have reported the mean performance over multiple runs in all our experiments. We kept this practice not only for our model but also in comparisons with other methods.
>
> For concerns about the experimental content and contribution (noted in **weaknesses**), please kindly refer to our detailed response to Reviewer #GeHM (Qs 1& 2), which covers these points.

---

> > ### Comment · Reviewer_L6VE · 2023-11-19
> > **Maintaining my original score**
> >
> > All of my concerns have been addressed, and I maintain my original score.

---

### Official Review · Reviewer_GeHM · 2023-11-01

**Soundness:** 2 fair
**Presentation:** 3 good
**Contribution:** 2 fair
**Rating:** 5
**Confidence:** 4

**Summary:**

This work introduces SimRec score, an anomaly score to improve detection of anomalies by combining the reconstruction score with a similarity score. The similarity score is computed via density estimations using a layer of RBF neurons in DNNs. Experimental results on various time-series anomalies demonstrate the performance of the proposed solution

**Strengths:**

- Significant value in improving anomaly detection methods by "small" modifications in current methodologies
- Experimental results support the claims of the paper
- Technical sound ideas

**Weaknesses:**

- Novelty is somewhat low
- Unclear experimental settings
- Missing related work, datasets, new benchmarks

**Questions:**

- Novelty is somewhat low

The idea seems technical sound as sub-components already exist. The technical novelty is somewhat low. For example, there is lack of preliminaries to help understand if something is brand new, but several of these questions are simple adaptation of the original equations applied to this new problem (e.g., k-means-type optimization in eq 3, RBF kernel already exists, combination of two scores is trivial, etc.)

The core idea of using sim to centroids has been applied many times (see work below), maybe not explicitly combined with the reconstruction error.

- Unclear experimental settings

The settings and parameters for baselines are not clear. Questions can be raised about fairness in comparisons when details from settings are omitted

- Missing related work, datasets, new benchmarks

The work focuses on a very narrow part of the literature (emerging due to the rise of DNN solutions) but definitely omits new progress on benchmarks, baselines, new evaluation measures [a,b].

New benchmarks, ~20 datasets, ~2000 timeseries, 10+ baselines
[a] TSB-UAD: an end-to-end benchmark suite for univariate time-series anomaly detection." Proceedings of the VLDB Endowment 15.8 (2022): 1697-1711.

New evaluation measures
[b] "Volume under the surface: a new accuracy evaluation measure for time-series anomaly detection." Proceedings of the VLDB Endowment 15.11 (2022): 2774-2787.

---

> ### Author Response · Authors · 2023-11-18
> **Response to Reviewer GeHM - Part (1/2)**
>
> We would like to thank you for your valuable comments and suggestions. We carefully addressed them point-by-point and made revisions to our manuscript accordingly.
>
> ### **Q1. Novelty is somewhat low:**
>
> We appreciate the opportunity to further clarify and underscore the contributions of our work, considering your concerns regarding the novelty of our methodology:
>
> - **Identification of a New Problem (Addressing a Specific Gap):** Our research has identified and addressed a previously unexplored challenge in anomaly detection models based on reconstruction error — specifically, their limited effectiveness in detecting subtle anomalies. Uncovering this issue is an important contribution to the field.
>
> - **Innovative Application of Existing Methods:** We agree with your point regarding the established nature of methods like the RBF, and we have already acknowledged this in the introduction of our paper. However, one of the novelties of our work lies in the specific application of these methods to address a specific problem. The combination of the RBF similarity score with traditional reconstruction error, while conceptually straightforward, is a strategic choice. This is not just a trivial combination but a carefully considered approach to enhance the sensitivity and robustness of anomaly detection for subtle anomalies (addressing the gap).
>
>     As you mentioned, this combination has not been explicitly explored in the literature, and our experimental results (provided by detailed comparative analysis) validate the effectiveness of our approach in addressing the identified gap.
>
>     We have updated the paper to elaborate more on how the combination of RBF similarity score and reconstruction error is not trivial and it has specific advantages - This claim is now even more supported by empirical results (using the new evaluation metrics suggested by you).
>
> - **Innovative Density Loss Term:** A key innovation in our approach is the introduction of a novel density loss term in the loss function, specifically designed for training the RBF kernel in this context. This term is detailed in the context of Formula 4 in our paper. We present the term here for reference:
>
>     $$
>     \text{Density Loss} = \lambda \frac{1}{N} \sum_{i=1}^{N} \frac{1}{T} \sum_{t=1}^{T} \log \left( \frac{1}{M} \sum_{m=1}^{M} (\mathbf{z}_{i,t})_m + \epsilon \right)
>     $$
>
>     This is a substantial technical contribution that enhances the effectiveness of our method and is not found in existing literature (as far as we were able to find).
>
> - **Efficiency of Practical Implications:** We demonstrate that next to using the novel density loss term, random initialization of the RBF parameters is as effective as more complex well-established initialization methods like K-means. This finding has significant practical implications, making our method more accessible and easier to implement in various settings.
>
>     &nbsp;
>
> In general, we have updated the paper to give a better overview of the contribution and novelty of the work. We hope that these points, along with the additional details and empirical evidence provided in the revised paper, address your concerns regarding the novelty of our work.
>
> &nbsp;
>
> ### **Q2. Unclear experimental settings:**
>
> We would like to clarify that detailed information about the settings and parameters is comprehensively provided in the appendix section of our paper. Additionally, we have made the full code of our experiments available to ensure transparency and facilitate reproducibility.
>
> To address your concern, we have now updated the key experimental details regarding the primary parameters, model architectures, and settings with a better overview of our experimental approach.
>
> We believe these updates will enhance the clarity of our experimental framework and ensure that our comparisons are both fair and easily replicable by other researchers in the field. We hope that these modifications address your concerns effectively.

---

> ### Author Response · Authors · 2023-11-18
> **Response to Reviewer GeHM - Part (2/2)**
>
> ### **Q3. Missing related work, datasets, new benchmarks**
>
> - **Incorporation of New Evaluation Metrics:** Following your recommendation, we have incorporated the new evaluation metric "Volume under the surface" from the referenced work [b]. This inclusion has not only validated our initial findings but has also strengthened the overall results and conclusions of our paper. We have updated our manuscript with the new results showcasing these results, demonstrating the added robustness and validity that this metric brings to our research. Your suggestion has significantly contributed to enhancing the quality of our work. (Due to space limitations, we only provide the new metrics here. See Table 1 from the updated paper for the complete results)
>
> **Table 1**. Mean performance (std). Initialization methods are denoted as (R) for Random and (K) for K-means.
>
> | Models                  | (SMD) VUS-ROC | | (SMD) VUS-PR | | (MSL) VUS-ROC | | (MSL) VUS-PR | | (PSM) VUS-ROC | | (PSM) VUS-PR |
> |-------------------------|---------------|-|--------------|-|---------------|-|--------------|-|---------------|-|--------------|
> | LSTM                    | 61(0)         | | 55(0)        | | 72(1)         | | 61(0)        | | 72(1)         | | 61(0)        |
> | SimRec-LSTM (R)         | **65(2)**     | | **60(3)**    | | **78(1)**     | | **65(2)**    | | **78(1)**     | | **65(2)**    |
> | SimRec-LSTM (K)         | **66(2)**     | | **61(2)**    | | **80(0)**     | | **66(2)**    | | **80(0)**     | | **66(2)**    |
> |                         |               | |              | |               | |              | |               | |              |  <!-- This is the empty row -->
> | Transformer             | 59(2)         | | 52(3)        | | 70(2)         | | 59(2)        | | 70(2)         | | 59(2)        |
> | SimRec-Transformer (R)  | **64(3)**     | | **57(3)**    | | **80(1)**     | | **64(3)**    | | **80(1)**     | | **64(3)**    |
> | SimRec-Transformer (K)  | **66(2)**     | | **61(4)**    | | **79(2)**     | | **66(2)**    | | **79(2)**     | | **66(2)**    |
>
> - **Benchmark Recommendations:** We understand your point regarding the inclusion of more diverse benchmarks, such as the TSB-UAD suite for univariate time-series anomaly detection. While we acknowledge the value of such comprehensive datasets, our research has targeted more challenging datasets. This is why we selected the multivariate time series datasets which are widely acknowledged and used in this field. We believe that focusing on these datasets allows us to show the effectiveness of our method in more complex, real-world scenarios. Other research has shown that simpler models can often effectively address anomaly detection in univariate datasets (refer to [a1]). However, our work aims to tackle the subtleties and complexities inherent in more challenging benchmarks. We appreciate your suggestion and will consider exploring a broader range of datasets, including univariate ones, in our future works.
>
> - **Expansion of Related Work:** We have updated the related research section in the introduction of our paper to include recent works and baselines in the field of anomaly detection. This addition ensures that our paper provides a comprehensive overview of the recent methods.
>
> - **Clarification on the Scope of the paper:** While acknowledging the variety of innovative approaches in anomaly detection, our research specifically focuses on addressing issues within Reconstruction Error (RE) based methods. Therefore, we have chosen baselines (for applying SimRec score) that are directly relevant to this focus. For instance, STAD-GAN [a2] is based on reconstruction error anomaly score, but a baseline like OmniAnomaly [a3], although effective, diverge from our core research area as they are based on different principles such as reconstruction probability, distance-based methods, or association discrepancy.
>
>     We believe that our focus on enhancing RE-based anomaly detection is a significant contribution to the field (both fundamentally and in terms of competitive performance compared to the recent SOTA method). Also, by identifying and addressing a specific limitation in these methods, our work opens up new opportunities for future research. For example, the integration of RBF kernels, as proposed in our method, could potentially benefit other baseline models.
>
> References:
> - [a1] Schmidl, Sebastian, et al. "Anomaly detection in time series: a comprehensive evaluation." Proceedings of the VLDB Endowment 15.9 (2022): 1779-1797.
> - [a2] Zhang, Zhijie, et al. "STAD-GAN: Unsupervised anomaly detection on multivariate time series with self-training generative adversarial networks." ACM Transactions on Knowledge Discovery from Data 17.5 (2023): 1–18.
> - [a3] Su, Ya, et al. "Robust anomaly detection for multivariate time series through stochastic recurrent neural network." Proceedings of the 25th ACM SIGKDD International Conference on Knowledge Discovery & Data Mining. 2019.

---

> > ### Comment · Reviewer_GeHM · 2023-11-20
> >
> > Thanks for addressing my concerns and clarifying the novelty aspects. With the new eval measures, I think the paper is much stronger now. You should make an additional effort to include many more baselines (TSBUAD identified 10 SOTA methods for example). As other reviewers point out, it's as if you mainly compare against a very narrow collection of methods, ignoring literature from data mining, databases, signal processing, and other areas. No matter what I increase my score but more SOTA methods, more datasets, can strengthen even more the paper.

---

> > > ### Author Response · Authors · 2023-11-22
> > > **New Response to Reviewer GeHM**
> > >
> > > Thank you for your feedback and for acknowledging the improvements made to our paper. We appreciate your suggestion to include a broader range of SOTA methods for comparison.
> > >
> > > We understand the value of comprehensive comparisons in demonstrating the effectiveness of new methods. However, we would like to clarify our rationale for the current scope of comparisons in our paper:
> > >
> > > - **Focus on Foundational Effectiveness of SimRec:**
> > >  As already mentioned, the primary goal of our paper is to demonstrate the foundational effectiveness of SimRec for anomaly detection. The comparisons in our paper are strategically chosen to showcase the impact of SimRec. The inclusion of a wide range of SOTA methods, while valuable and providing additional context, may not necessarily enhance the central message of our work and might shift the focus from the core contribution of our paper - the effectiveness of SimRec method. We believe that the results presented in Table 1 are the core of our paper, illustrating the significant impact of the SimRec method, which is our main contribution.
> > >
> > >
> > > - **Competitive Performance with Existing SOTA:** We have shown that our results are competitive with a recent SOTA method, Anomaly Transformer. This model has already demonstrated superiority over the various SOTA baselines mentioned in your suggested paper (Available in their publication). By comparing with Anomaly Transformer, we aimed to provide a clear and direct overview of SimRec’s effectiveness, although outperforming the SOTA was not the primary goal of our paper.
> > >
> > > - **Scope and Directness:** To maintain the clarity and directness of our message, we chose not to pursue a broad comparison with various SOTA methods. This decision was made to keep the focus on illustrating SimRec's impact rather than pursuing superior performance through tailored architectures or optimization parameters.
> > >
> > > In summary, while we recognize the potential benefits of a broader range of comparisons, we decided to focus our efforts on demonstrating the novel contribution of the SimRec method. We believe this approach provides clear and valuable insights to the readers in the field of anomaly detection.
> > >
> > > Thank you for your understanding and support, and we hope this response clarifies our research approach and addresses your concerns. We are excited about the possibility of our work contributing significantly to the venue.

---

### Meta-Review · Area_Chair_MJp3 · 2023-12-06

**Metareview:**

The manuscript proposes a new score to complement reconstruction error for anomaly detection from multi-variate time-series sensor data. The method is evaluated on 3 data sets, and performance results support the author's claims.

Although a clever contribution on scoring strategies for time-series anomaly detection, the novelty is minimal (All Reviewers)
The manuscript lacks sufficient baselines, and the experiments are conducted only on LSTM and Transformers (All reviewers)
Reviewer L6VE had suggested a reconstruction error graph of anomaly transformer using the same threshold selection -  to see the effect of threshold selection and the authors have not addressed this

**Justification For Why Not Higher Score:**

Although an important contribution, the novelty is a bit weaker considering the RBF is a well-explored field.

Although a clever contribution on scoring strategies for time-series anomaly detection, the novelty is minimal (All Reviewers)
Reviewer L6VE had suggested a reconstruction error graph of anomaly transformer using the same threshold selection -  to see the effect of threshold selection and the authors have not addressed this

**Justification For Why Not Lower Score:**

The proposed score for anomaly detection can be a new complement to existing reconstruction error, and may be of interest.

---

### Decision · Program_Chairs · 2024-01-16

Reject